# Prickly Pear Seed Oil Extraction, Chemical Characterization and Potential Health Benefits

**DOI:** 10.3390/molecules26165018

**Published:** 2021-08-19

**Authors:** Ghanya Al-Naqeb, Luca Fiori, Marco Ciolli, Eugenio Aprea

**Affiliations:** 1Center Agriculture Food Environment, University of Trento, Via E. Mach, 1, 38010 San Michele all’Adige, TN, Italy; luca.fiori@unitn.it (L.F.); marco.ciolli@unitn.it (M.C.); eugenio.aprea@unitn.it (E.A.); 2Department of Food Sciences and Human Nutrition, Faculty of Agriculture Foods and Environment, University of Sana’a, Sana’a 009671, Yemen; 3Department of Civil, Environmental and Mechanical Engineering, University of Trento, Via Mesiano 77, 38123 Trento, TN, Italy; 4Department of Food Quality and Nutrition, Research and Innovation Centre, Fondazione Edmund Mach, Via E. Mach, 1, 38010 San Michele all’Adige, TN, Italy

**Keywords:** prickly pear seed oil, extraction, chemical characterization, biological activity

## Abstract

Prickly pear (*Opuntia ficus-indica* L.) is a member of the Cactaceae family originally grown in South America, and the plant is now distributed to many parts of the world, including the Middle East. The chemical composition and biological activities of different parts of prickly pear, including cladodes, flowers, fruit, seeds and seed oil, were previously investigated. Oil from the seeds has been known for its nutritive value and can be potentially used for health promotion. This review is an effort to cover what is actually known to date about the prickly pear seeds oil extraction, characteristics, chemical composition and potential health benefits to provide inspiration for the need of further investigation and future research. Prickly pear seeds oil has been extracted using different extraction techniques from conventional to advanced. Chemical characterization of the oil has been sufficiently studied, and it is sufficiently understood that the oil is a high linoleic oil. Its composition is influenced by the variety and environment and also by the method of extraction. The health benefits of the prickly pear seed oil were reported by many researchers. For future research, additional studies are warranted on mechanisms of action of the reported biological activities to develop nutraceutical products for the prevention of various chronic human diseases.

## 1. Introduction

*Opuntia ficus-indica* L., which is known as a prickly pear, belongs to the *Cactaceae* family and has spread widely throughout the world. The prickly pear plant is used nowadays in many forms, including food, feed, health and nutrition and also as prepared products including cosmetics, tea, jam, juice and oil extracted from the seeds. Health benefits and biological activities have been reported for different parts of the prickly pear plant, such as cladodes, fruit, seeds and seed oil. The entire fruit of prickly pear contains 9–10% of seeds [1,2]. The prickly pear seeds showed to contain polysaccharides, cellulose and hemicelluloses [3]. Many studies have suggested prickly pear seeds as a new source of fruit oils [4,5]. The extracted oil presented 5–15.5% of total seed composition [1,2,6,7].

The growing demand for nutraceuticals and functional foods is paralleled by an increased effort in developing natural products for the prevention or treatment of human diseases. According to several studies, prickly pear seed oil (PPSO) contains high values of important nutrients that make the PPSO an excellent candidate for inclusion in food and healthy products; however, more research is needed to establish new pharmacological possibilities for its further future development and application. Unsaturated fatty acids are the main component of the oil, accounting for up to 80.9%. However, the main unsaturated fatty acids are linoleic and oleic acids [6,7,8].

The PPSO is reported to have several biological activities, including antioxidant in vivo [9] and in vitro [10,11], antimicrobial [12,13], antidiabetic [3,6,14,15], lipid-lowering [14], in vitro anticancer [12,15], anti-inflammatory [9,16], anti-ulcer [17] and UV radiation protection of human dermic fibroblasts reducing the cell death [18]. Based on the literature, the oil content in prickly pear seeds is considered low compared to other seeds and the extracted oil is commercialized mainly for cosmetics due to the high price resulting from the time-consuming and laborious process of production [19]. Analysis and physicochemical properties of the PPSO showed that it is edible and suitable for human consumption.

This paper aims to review the scientific literature systematically and provide a comprehensive summary of prickly pear seeds oil, its extraction, characteristics, chemical composition and pharmacological activities.

## 2. Prickly Pear Seed Oil Extraction and Yield

Due to the multiple health benefits of PPSO, an extraction process to preserve the quality of PPSO is of prime importance and thus should receive continuous attention from researchers. Based on the previous studies, PPSO was extracted using conventional extraction methods, such as Soxhlet and cold press, and alternative advanced extraction methods, such as supercritical carbon dioxide. Different extraction processes were also applied in PPSO extraction, including maceration, autoclave, microwave and ultrasound-assisted extraction. However, conventional methods using Soxhlet extraction are the most reported methods in the literature, while limited studies have reported the used of advanced extraction methods. Based on the previous studies, the oil yield was found to be varied depending on the prickly pear cultivar in the first place and second on the extraction process. Several factors, including geographical origin, harvest period, fruit ripeness and extraction solvents are also included [1,4,7,11].

### 2.1. Conventional Extraction Methods

Conventional extraction methods, including solvent extraction and mechanical extraction, were used to extract PPSO in the reported studies. Generally, the majority of scientific works deal with solvent extracted oils that give a better oil yield. Different solvents were used in different studies, and the most commonly used are hexane, petroleum ether, ethyl acetate and chloroform-methanol (2:1). From the different studies reported, three steps in solvent extraction are applied—seed preparation, oil extraction, and solvents evaporation. Conventional seed preparation includes separating the seeds from the fruit and then cleaning, drying and grinding, as presented in Figure 1.

Solvent extraction with the Soxhlet system was the most frequent extraction method in the reported studies. The solvent used for PPSO extraction has an influence on PPSO yield: in this context, a study that compared PPSO extraction yields using two different solvents 2-methyloxolane (2-MeO) and *n*-hexane using the Soxhlet system for 8 h indicated that PPSO yield obtained was different in the two solvents. The yield with 2-MeO was higher (9.55 ± 0.12 g/100 g) compared to the yield obtained with *n*-hexane (8.86 ± 0.25 g/100 g) [22]. In addition, oil yield varies depending on prickly pear varieties and locations. Kolniak-Ostek et al. [23] found that the PPSO yield extracted from seeds of seven Spanish prickly pear cultivars, with diethyl ether using Soxhlet apparatus for 3 h, varied from 2.61% for “Nalle” to 7.69% for “Nopal ovalado”. In this line, the PPSO yield was found to be 5.4–9.9 g/100 g when the oil was extracted from 17 different origins in Morocco with *n*-hexane for 8 h using the Soxhlet apparatus [4]. In addition, the PPSO yield extracted with petroleum ether in a Soxhlet extractor for 6 h was found to be 5.0–14.4% depending on the different localities in Turkey [24].

Moreover, the yield of PPSO varies depending on the extraction time applied from the same location, for example, when PPSO was extracted from prickly pear seeds from Algeria with *n*-hexane using the Soxhlet system, the yield of PPSO was 10.45 ± 0.10% when the extraction was carried out for 18 h [25], and the yield was 7.3–9.3% when the extraction was carried out for 9 h [26]. In contrast, the oil yield in Tunisian prickly pear seeds was approximately similar when the variation of the extraction time applied was low, and when the PPSO was extracted with *n*-hexane using the Soxhlet extractor for 10 h, the yield was reported to be 11.75% [1]. A similar PPSO yield (11%) was reported in another study by [27] of Tunisian PPSO extracted with *n*-hexane in a Soxhlet extractor for 9 h. However, a different Tunisian PPSO yield was reported between the PPSO extracted from seeds of wild prickly pear (10.32%) compared to cultivated prickly pear (8.91%) [28].

The PPSO yields were also affected by the different geographical locations; the PPSO yield extracted from prickly pear in Greece with *n*-hexane using the Soxhlet apparatus for 4–6 h was 5.4 ± 0.5% [7]. In addition, a PPSO yield of 16.2 g/100 g was obtained when the dried prickly pear seeds powder originated from Egypt extracted with methanol in a Soxhlet apparatus for several hours [29]. However, the approximate analysis of the prickly pear seed from Saudi that was extracted with petroleum ether in a Soxhlet extractor was 17.2% total fat [30].

Maceration extraction with solvents is another method that was used for extracting PPSO. Different solvents were used for PPSO extraction, and the PPSO yield varied depending on solvents used and varieties of prickly pear. The oil yield of PPSO was found to be 11.83% for the green variety and 6.89% for the red in the oil extracted using maceration extraction by *n*-hexane, whereas the PPSO yield was not different among the green and red varieties when the oil was extracted with either ethanol or ethyl acetate, and the yield was 10% in both varieties [13]. In addition, chloroform and methanol in a ratio of 2:1 were used to extract total lipid, and the yield results ranged from 4.09 to 8.76% as reported [31].

The effects of different factors on PPSO yield and fatty acid composition were reported by De Wit et al. [32]. Total lipid was extracted using chloroform and methanol (2:1, *v*/*v*) with additional butylated hydroxytoluene as an antioxidant to the chloroform at a concentration of 0.001%. Oil content significantly varied among different cultivars, location, and season, where the average oil content varied from 5.8 to 6.7% between locations, 5.8–6.67% between seasons, and 3.45–8.23% among cultivars. Moreover, PPSO extracted from prickly pear seeds originated from Yemen, using maceration extraction with different solvents in the same conditions, indicated that chloroform and methanol in a ratio of 2:1 showed higher yield (7.76 ± 0.43%), following by petroleum ether (6.1 ± 0.41%) and hexane (5.0 ± 0.36) [6]. In this line, Ramadan and Mörsel [4] reported the extraction of the PPSO from cactus pear obtained from a local market in Berlin, Germany. Powder seeds were homogenized in methanol for 1 min in a blender, chloroform was added, and homogenization was continued for a further 2 min. The mixture was filtered, and the solid residue was re-suspended in chloroform/methanol (2:1, *v*/*v*) and homogenized. Then, the mixture was filtered again, and combined filtrates were cleaned. The yield was reported as 9.9 g oil /100 g seeds. 

Mechanical extraction using an electric powered screw press was used for extracting PPSO [19]. The extraction process was followed by centrifugation of the oil for 15 min at 3000 rpm to separate it from the sediment; no yield of the extracted oil was reported in this study. In another study, the extraction of PSSO was obtained by cold pressing the dried seeds using a mechanical oil press [9]. However, in comparison to the study reported by Regalado-Rentería et al. [18], the PPSO was extracted using a cold press and maceration methods with *n*-hexane for 2 h, the cold press method resulted in lower PPSO yield ranging from 0.51 to 6.1 g/100 g, whereas the maceration-percolation method resulted in a PPSO yield in the range 6.2–15.54 g/100 g.

### 2.2. Innovative Techniques

Few studies have reported the use of innovative methods such as supercritical fluid extraction, microwave extraction and ultrasound extraction in PPSO extraction. However, it is expected that they will be widely used for PPSO in the near future due to the importance of the health-promoting qualities of PPSO.

In addition, the extraction of PPSO using supercritical fluid extraction at different pressures ranging from 100 to 500 bar and different temperatures of 35–50 °C at a carbon dioxide flow rate of 50 g/min was reported with a yield of 6.5% [33]. In this context, a supercritical fluid extraction method in comparison with the Soxhlet method was applied to PPSO extract from two Tunisian prickly pear seed types—the spiny (wild) and thornless (cultivated) [28]. Supercritical fluid extraction was carried out at different temperatures of 35, 40, 45, 50, 60, and 70 °C, fixed pressure at 180 bar, and 15 mL/s of CO_2_ flow rate for 135 min. The results of the optimized temperature in supercritical fluid extraction showed that a higher yield was obtained at 40 °C. Results obtained from this study show a significantly higher yield of 10.32% (wild) and 8.91% in Soxhlet with *n*-hexane compared to supercritical fluid extraction yielding 3.4% (spiny) and 1.94% (thornless) [28]. Green cactus pear seed oil was extracted using ultrasounds with an obtained yield of 3.75–6% [11]. The recovered extract obtained with supercritical fractionation of the defatted seeds of the prickly pear from Algerian *Opuntia ficus*-*indica* was rich in antioxidants catechin, epicatechin, and ferulic acid, which are known to be good for human health [34].

In situ microextraction of seed oil components using a microwave method is an optional method of quantifying metabolites present in the seed when only a small sample is available. Only one study has reported using this method to extract PPSO from eight variants of prickly pear in Mexico, which were mainly used for the recovery of phytochemicals from the oil, including squalene, β-sitosterol and γ-tocopherol [18].

Moreover, only one study has reported the extraction of prickly pear essential oil from two Sicilian cultivars of red and yellow fruits. The essential oil yield was 3% in the seeds of yellow fruits, which was higher than the yield obtained from seeds of red fruits, 0.63% [35]. Different extraction methods, solvent, extraction time and yield of PPSO are shown in Table 1.

## 3. Characteristics of Prickly Pear Seed Oil

Few studies have reported the physicochemical properties of PPSO, including peroxide value, iodine value, acidity index, refractive index and saponification value. Physicochemical characteristics of PPSO, as reported in the literature, are summarized in Table 2.

Recently, the chemical and physical parameters of PPSO extracted by two different solvents—2-MeO and *n*-hexane—were reported [22]. The oil extracted by 2-MeO showed a higher acidity index (3.02 ± 0.5, g/100 g) compared to the *n*-hexane extracted oil (1.26 ± 0.5 g/100 g). The peroxide value was found to be high in the two different solvents (3.5 ± 1.5 in hexane oil and 8.6 ± 2.5 in 2-MeO), which can be an indicator for primary fat oxidation [37]. Additionally, the physicochemical characteristics of PPSO extracted with *n*-hexane using a Soxhlet apparatus were not different from the study of Gharby et al. [22], where the acid value was 1.82 ± 0.01, iodine value was 93.45 ± 0.22 (g of I2/100 g of oil), saponification value was 177.10 ± 0.05 (mg of KOH/g of oil), density was 2.04 ± 0.05 at 20 °C (mass/volume), and the refractive index was 0.909 ± 0.01, with a yellow-brown color [26].

In this context, the physicochemical characteristics of PPSO extracted with *n*-hexane in a Soxhlet extractor showed PPSO with a density of 0.903 ± 0.002, the refractive and iodine indices reported as 1.475 ± 0.002 at 20 °C. 101.5 ± 1.0, respectively, while the saponification index (mg of KOH/g oil) was 169.0 ± 0.1 [27]. In addition, the physicochemical characteristics of PPSO extracted by first cold pressing did not differ much from other studies previously mentioned: the PPSO was found to have a greenish-yellow-colored liquid, a density of 0.905 ± 0.001 at 20 °C, an acid index of 1.952 ± 0.034, an iodine index of 108.52 ± 0.250 (g I2/100 g oil), a peroxide index of 2.230 ± 0.061 (meq O_2_/kg oil), a saponification index of 171.40 ± 0.430 (mg KOH/g oil), and a refractive index at 20 °C of 1.475 ± 0.001 [17]. However, the physical and chemical parameters of PPSO oil from Algeria extracted by cold pressing showed different results compared to other studies. The PPSO had an acidity of 21.2 ± 0.5 mg/g of oil (expressed as % of oleic acid), and the author explained the high acidity due to enzymatic hydrolysis during harvesting or the handling process. The peroxide value was found to be 12.0 ± 0.4 meq O_2_/Kg. Compared to other studies on PPSO, the peroxide value reported in this study was also higher [36]. The authors have explained the elevated peroxide value as a result of unsatisfactory conditions used during the oil preparation.

## 4. Chemical Characterization of the Prickly Pear Seed Oil

### 4.1. Fatty Acid Composition

Various researchers have reported the fatty acid composition of PPSO. Gas chromatography of the PPSO methyl ester was the main method for fatty acid composition. PPSO is characterized by a high level of unsaturated fatty acids (80–88%), among which are linoleic acid (49.3–78.8%), oleic acid (12.8–25.3%), vaccenic acid (4.3–6.3%) and linolenic acid (0.23–1.1%). The main saturated fatty acids are palmitic (9.3–14.3%) and stearic (2.2–4.3%) acid. The main fatty acid compositions of PPSO are summarized in Table 3.

From different reported studies, various factors have an influence on fatty acid content, including prickly pear variety, geographical location, methods and solvents used for oil extraction, cultivar, degree of maturity, and harvesting season. Recently, the fatty acid composition of the oil extracted by two different solvents using 2-MeO and *n*-hexane were found to be similar [22]; linoleic acid was the major fatty acid constituting up to 62%, followed by oleic acid (21%), linolenic acid (0.3%) with 84% counting as unsaturated fatty acid. Saturated fatty acid represented 16%, with palmitic acid counting as 11.70% and stearic acid 3.14% [22]. However, fatty acid content, especially linolenic acid and oleic acid, was found to be different depending on the extraction method. The fatty acids of PPSO were extracted from eight varieties of prickly pear from Mexico using two different methods—cold press and maceration [18]. The cold hydraulic press method showed slightly higher fatty acid contents when compared to the maceration method, where the oil analysis showed that linoleic acid was in the range of 66.5–76.1 and 60.5–78.8, followed by oleic acid in the range of 9.3–19.9 and 11.5–19.9 for hydraulic press and maceration methods, respectively.

Although, for most reported studies from different varieties and different locations, the main fatty acids were linoleic and oleic acids, and the content of fatty acid was different. Moreover, differences were found in the oil composition of the PPSO derived from various places. In this context, the fatty acids analysis of PPSO extracted from seven Spanish prickly pear cultivars showed differences between fatty acid content in different varieties, which were attributed to genetic factors [23]. The major fatty acid was linoleic (57.72–63.11%), followed by oleic acid (19.37–21.79%), linolenic acid (0.23–1.10%), palmitic acid (12.47–15.06%) and stearic acid (2.56–4.10%). In this regard, the analysis of fatty acid of PPSO from different places in Morocco showed unsaturated fatty acids of 83.0%, where the major fatty acids were linoleic acid (60.2–64.6 g/100 g), oleic acid (18.2–22.3 g/100 g), linolenic acid (0.3 g/100 g), and palmitic acid (11.6–12.4 g/100 g) [2]. In addition, fatty acid content was varied due to time collection from different locations and provinces of Turkey [24]. The unsaturated fatty acids included mainly linoleic (49.3–62.1%), oleic acid (13.0–23.5%) and vaccenic acids (5.0–6.3%). Saturated fatty acids mainly include palmitic acid contents of seed oils (10.6–12.8%) and stearic acid (3.3–5.4%). 

Moreover, fatty acid content also differed among wild and cultivated prickly pear grown in Cyprus [16], and the main compounds of wild oil were linoleic acid that accounted for 55.9%, oleic acid 17.6%, palmitic acid 12.4% and elaidic acid 4%, while cultivated oil contains linoleic acid of 60.1%, oleic acid 15.6%, palmitic acid 12.3% and elaidic acid 4.1%. Among the two typologies, cultivated contained more linoleic acid compared to the wild species. In this regard, De Wit et al. [32] stated that variation in rainfall and cultivar showed a significant impact on PPSO fatty acid composition comparing 42 Spineless Burbank *Opuntia ficus-indica* cultivars in South Africa. The major fatty acids were linoleic acid, which varied from 56.86 to 65.21%, oleic acid ranging from 16.44 to 22.51%, palmitic acid content ranging from 12.72 to 16.05% and stearic acid ranging from 2.21 to 3.39%.

In addition, the fatty acid profiles of two forms of Tunisian *Opuntia ficus-indica* seeds in the wild (spiny) and (cultivated) thornless were reported [28]. Linoleic acid was the major unsaturated fatty acid, representing 57.60% in PPSO of wild variety and 59.98% in PPSO of cultivated variety extracted by supercritical fluid extraction, and a similar amount of linoleic acid was found in the PPSO of cultivated and wild varieties extracted using *n*-hexane in Soxhlet apparatus. Oleic acid in the wild and cultivated varieties were different based on the extraction method. Oleic acid was 22.31 and 22.40% in wild and cultivated varieties, respectively, of PPSO extracted by supercritical fluid extraction and 25.28 and 20.58% in wild and cultivated varieties, respectively, in PPSO extracted using *n*-hexane with Soxhlet apparatuses.

Geographical location also has an impact on fatty acid content of PPSO, especially the content of linoleic acid, even though the PPSO is extracted with the same method. A study on the fatty acid profile of Algerian PPSO extracted by cold press indicated that Algerian PPSO contains linoleic acid (49.7–56.1%), oleic acid (15.6–19.3%), vaccenic acid (4.30%) and α-linolenic acid (0.24%). Saturated fatty acid mainly includes palmitic acid (10.08%) and stearic acid (2.97%) [36]. Fatty acid composition of Egyptian PPSO extracted with methanol in a Soxhlet apparatus showed that linoleic acid was found to be the major unsaturated fatty acid (54.03%), followed by oleic acid (22.41%) and linolenic acid (0.63%); whereas, the major saturated fatty acid was palmitic acid (17.11%) and stearic acid (3.5%) [29]. However, the fatty acid content of Tunisian PPSO extracted using the cold press method as well was found to contain higher linoleic acid (61.6%) and higher oleic acid (21.18%). Linolenic acid was counted as 0.2%. Furthermore, the saturated fatty acids were higher, representing 16%, which included palmitic acid (12.2%) and stearic acid (3.3%) [17]. In this context, the fatty acid composition of PPSO from Sicilian yellow fruit extracted from the seeds using cold pressing was in the range of the reported studies. Unsaturated fatty acids represent 83.79%, with 58% linoleic acid content, 18% oleic acid, 6.29% vaccenic acid, and 0.3% arachidic acid. Saturated fatty acids counted for 16.31%, where palmitic acid content of 12% and stearic acid of 4% were the major ones [38]. The results from this study also reported that the oil obtained in Sicily contained significant amounts of some unsaturated fatty acids, which have been known for their health-benefiting properties, including trans-13-octadecenoic, 7Z, 10Zhexadecadienoic, and gadoleic acid.

Geographical location with a different extraction method has an effect on the fatty acid composition; in this regard, analyses of fatty acid composition in PPSO from prickly pear seeds cultivated in Yemen extracted using the maceration method with *n*-hexane was investigated [6] and unsaturated fatty acids were found to be the majority of fatty acid content, representing up to 81% in oil with linoleic acid content of 57.5% and oleic acid content of 22.30%. On the other hand, the major saturated fatty acids of PPSO were palmitic (14%) acid and stearic acid (3%). In this regard, a high percentage of unsaturated fatty acids was reported in the PPSO from Tunisia extracted using *n*-hexane with a Soxhlet system in which linoleic acid was the major component making up 70%, followed by 17% oleic acid. Saturated fatty acid represents 12.4%, and palmitic acids (9%) and stearic (3%) were the major saturated fatty acids [15]. Moreover, it was found that the fatty acid content of PPSO varied due to different locations and provinces of Turkey, and also, based on the time collection, the content of linoleic ranged from 49.3 to 62.1%, oleic acid from 13.0 to 23.5% and vaccenic acids from 5.0 to 6.3%. Saturated fatty acids mainly include palmitic acid contents of seed oils of 10.6–12.8% and stearic acid 3.3–5.4% [24].

De Wit et al. [31] indicated significant differences in most of the fatty acid compositions between locations. In addition, seasonal differences affected most of the fatty acid profiles of PPSO significantly except for C16:1c9, C18:2c9,12 and C20:1c11. Rainfall showed large variation across seasons for fatty acid content. The study also has indicated that oleic acid was the only fatty acid that was significantly influenced by cultivar × location and location × season interactions, with the content ranging between 16.14 ± 0.08 and 22.51 ± 1.01.

Moreover, fatty acid composition of PPSO indicated that the linoleic acid was the major fatty acid (53.5 ± 4.9%) followed by oleic acids (18.3 ± 1.6%) and a-linolenic acids (2.58 ± 0.165), whereas palmitic acid (20.1 ± 2.3%) and stearic acid (2.72 ± 0.1%) were the main saturated fatty acids [4]. We observe that this study has reported the highest content of linolenic acids and palmitic acid compared to all the previous studies. 

### 4.2. The Content of the Phytosterols and Tocopherols of Prickly Pear Seed Oil

Prickly pear seed oil contains other valuable compounds, such as sterols and tocopherols, that have been known and approved scientifically to decrease or alleviate human diseases, such as atherosclerosis, diabetes and cancer [39]. The content and composition of the phytosterol is considered as a fingerprint of the oil, and it can be used for authenticity or detecting adulterations [40]. β-sitosterol and campesterol were found to be the major phytosterols compounds. The composition of phytosterols and vitamin E (%) of total sterol and total vitamin E reported in the literature is summarized in Table 4.

Recently, the phytosterol and tocopherol compositions of Moroccan PPSO were reported [22]. Total sterol content was 1110.5 ± 2.5 and 1020.1 ± 7.54 mg/kg oil in the 2-MeO and *n*-hexane, respectively, and the major components were found to be β-sitosterol (75–82%) followed by campesterol (11.6–12.3%). Other phytosterols reported in this study, including cholesterol, Δ5-avenasterol, Δ7-stigmasterol, Δ7-avenasterol, were found to be the minor sterols. The study also indicated that no significant differences in vitamin E content and its compositions in PPSO extracted by the two solvents, hexane and 2-MeO. The tocopherol content was 78.75 and 77.85 mg/100 g in the PPSO extracted with *n*-hexane and 2-MeO, respectively. γ-Tocopherol was the major form present with 68.4 and 67.46 mg/100 g), respectively, in the two solvents, followed by the δ-tocopherol (7.5 and 8.26 mg/100 g) and α-tocopherol (1.75 and 1.95 mg/100 g), whereas the β-tocopherol in PPSO-extracted oil with both solvents was absent. However, less total sterol in PPSO from Morocco was reported [41], where the PPSO had a total phytosterol content of 832.90 mg/kg oil, and β-sitosterol counted as 74.71% of total sterol, followed by campesterol with 10.20%. Other phytosterols were found in small quantities, including Δ7-Stigmasterol, Δ7-Avenasterol and cholesterol. In addition, total tocopherol was found to be 81.5 mg/100 g, where γ-tocopherol was found to be the major form of tocopherols in PPSO counting for 98.45%, α-tocopherol counted for 1.09%, and δ-tocopherol counted for 0.46%.

From previous studies, geographical location, varieties and method of oil extraction were found to have an influence on the phytosterol and vitamin E composition. In this context, the sterol analysis of PPSO extracted from Algerian prickly pear seeds using cold press methods was reported [36]. β-sitosterol counted for 387.44 ± 3.04 mg/100 g oil followed by 47.04 ± 0.02 stigmastanol, 21.65 ± 0.09 campesterol and 11.26 ± 0.51 stigmasterol. However, the phytosterol composition of Tunisian PPSO extracted firstly by cold pressing showed 81% β-sitosterol of total phytosterol followed by 11% campesterol. The analysis also indicated the presence of campesterol Δ-7-avenasterol, stigmasterol, clerosterol, Δ-7-stigmastenol, Δ-5-24-stigmastadienol in minor amounts [17]. Total tocopherol was 86.3 ± 1.16 mg/100 g, where γ-tocopherols counted for 92%, δ-tocopherol counted for 6.24% and α-tocopherol counted for 1.33% of total tocopherol [17]. The findings reported [2] on sterol composition of PPSO from Tunisia indicated that β-sitosterol was the main phytosterol in PPSO representing 79.1% of the total sterol. Other reported sterols are campesterol (10.0% of the total sterols), delta-5-avenasterol (5.1%), and stigmasterol (2.5%).

The phytosterols content of PPSO from Tunisia extracted using a Soxhlet extractor with *n*-hexane was also reported as 16.06 ± 0.28 g/kg [27]. β-sitosterol was found to count for 72%, and Δ5-avenasterol and Δ5-avenasterol counted for 4.72 and 5.10% of the total sterol content, respectively. Vitamin E content was also analyzed in Tunisian PPSO and was found to be 447.380 ± 0.140 mg/kg, where γ-Tocopherol showed 94.12% of the total vitamin E content [1]. Two new sterols—schottenol (11.6 mg/100 g oil) and spinasterol (14.44 mg/100 g oil)—were presented in PPSO from Morocco [42], where total sterol was found to be 900.68 mg/100 g oil. Schottenol and spinasterol Structures are shown in Figure 2.

Moreover, differences were found in the vitamin E content and composition of the PPSO derived from various locations and the time collection from Turkey [24]. The total amount of tocopherols ranged between 3.9 and 50.0 mg/100 g. In addition, the β-tocopherol content of Opuntia seed oils varied between 9 and 50.0% in different oils from different locations and times of the seed collection.

### 4.3. Polyphenols, Flavonoid, Carotenoid, and Chlorophyll Contents of Prickly Pear Seed Oil

Other than sterols and tocopherols, the antioxidant activity may be partly due to some compounds, e.g., phenolics. Few studies have reported the polyphenol content in PPSO, but recently, total polyphenol content (TPC) in Algerian PPSO extracted by the cold press method was reported to be 55.82 ± 3.84 mg of gallic acid equivalents/100 g of oil [36]. A similar finding of TPC was reported where the TPC content of Greece PPSO extracted using *n*-hexane with a Soxhlet apparatus was 55.1 ± 0.300 mg of gallic acid equivalents/100 mL oil [7]. Higher TPC content of 416 mg of gallic acid equivalents/100 g in Algerian PPSO was reported [43]. However, TPC content of Moroccan PPSO extracted by the cold press method has shown to be 26.5 gallic acid equivalents/g oil [17]. 

The characterization of phenolic compounds of PPSO extracted using cold press from six different locations in Morocco was recently reported [44]. As shown in Figure 3, seven different phenolic compounds were identified belonging to three families, which are hydroxyl cinnamic acid derivates (*p*-coumaric acid, *p*-coumaric acid ethyl ester, ferulic acid), hydroxyl cinnamaldehyde derivates (furaldehyde) and hydroxybenzaldehyde derivates (4-hydroxy benzaldehyde, vanillin, syringaldehyde), where vanillin, syringaldehyde, and ferulaldehyde were the higher compounds. The structural formulas of phenolic compounds identified in PPSO are presented in Figure 3.

Moreover, recently, the authors of [23] reported on the determination of polyphenolic compounds from seeds of seven Spanish prickly pear cultivars, indicating that total phenolic compounds were significantly different depending on the seed varieties and ranged from 34.07 to 266.67 mg/kg. A total of 21 metabolites were also identified from the PPSO, where phenolic compounds and flavonols were the two major classes identified. In addition, the study of [23] reported the effect of seed roasting on the identified compounds, where some phenolic compounds were increased after a roasting process that was carried out at 110 °C at different times (10–40 min).

Only a few studies have evaluated the flavonoid content of PPSO. The flavonoid content was 1.5 ± 0.1 and 2.6 ± 0.2 mg quercetin eq/100 g in [36]. However, the authors of [17] indicated a flavonoid content of 3.1 mg quercetin eq/g oil, carotenoid content of 10.520 mg/kg oil, which was related to lutein, and total chlorophyll content of 4.57 mg/kg oil, which was related to pheophytin. The evaluation of carotenoid levels of PPSO was also reported [4], and it was restricted to β-carotene, which accounted for 0.05 ± 0.01 g/kg, as the authors owed the light-yellow hues of PPSO to the presence of the β-carotene. Quercetin and its derivatives and isorhamnetin derivatives have been identified in PPSO as well [23]. The chemical structures of pheophytin and lutein fractions are presented in Figure 4.

### 4.4. Prickly Pear Seed Oil Volatile Compounds

In previous studies, the available data regarding the volatile compounds in PPSO are limited since PPSO is considered an edible oil and suitable for human consumption. In recent years, the volatile compounds of the oils, including hydrocarbons, aldehydes, ketones, esters and acids, have gained interest due to their importance in the sensory quality of the edible oils assessment. The presence or absence of these compounds and their amounts depended on varieties and oil extraction methods. The main classes of volatile compounds of cactus seed oil reported in the literature are summarized in Table 5.

In this regard, the profile of volatile aroma-active compounds of PPSO has been reported [19]. The study has determined the profile of volatile compounds of PPSO extracted by the cold press method from five different regions of Morocco. Approximately 32 volatile compounds were reported in this study. The main volatile compounds identified were hexanal and 2-methyl propanal, with average amounts of 57.4 and 38.9 mg/kg, respectively. Additional compounds were 16.2 mg/kg of acetaldehyde, 10.9 mg/kg of acetic acid, 10.2 mg/kg of acetoin and 6.3 mg/kg of 2,3-butanedione. (The authors have attributed the flavor of the PPSO to the presence of these compounds). The study also reported how roasting cactus seeds at 110 °C could affect the composition of volatile compounds in the oil. Specifically, the concentration of compounds known as products from the Maillard reaction increased significantly with roasting time, such as furfural, furan, 2-butanone and 5-methyl furfural. (The study also indicated that roasting time is important for increasing the aroma-active compounds, which results in a greater intensity of the smell). In addition, the volatile compounds of PPSO from Greece [7] showed the identification of 121 volatile compounds using headspace solid-phase micro-extraction coupled to gas chromatography/mass spectrometry. According to this study, the volatile compounds were grouped in acids (2.70%), alcohols (9.13%), aldehydes (62.72%), esters (2.82%), hydrocarbons (5.06%), ketones (4.38%), and other compounds (12.71%). The most dominant volatile compounds were aldehydes, (among which) 2-propenal, pentanal, hexanal, (*E*)-2-hexenal, heptanal, (*Z*)-2-heptenal, octanal, 2-octenal, nonanal, 2,4-decadienal, and trans-4,5-epoxy-(*E*)-2-decenal recorded the higher proportions. Moreover, a study on the essential oil composition of seed oil from the Sicilian cultivars of *Opuntia ficus-indica* red and yellow fruits (Sanguigna and Surfarina cultivars) was reported [35]. About 41 compounds have been identified, and the red seeds contain mainly hydrocarbons (38.5%), fatty acids and derivatives (31.9%) and terpenes (12.4%), while the yellow seeds contain the highest presence of fatty acids and derivatives (68.9%) and terpenes (10.9%).

## 5. Potential Health Benefits of Prickly Pear Seeds Oil

The potential health benefits of PPSO have attracted increasing attention. The authors of a number of studies have suggested that the primary benefit of PPSO is due to its polyunsaturated fatty acids, sterols and antioxidant activity. The biological activity of PPSO reported in the literature, conducted mainly in animal studies, showed that the PPSO exerts several health benefits. 

### 5.1. Antioxidant Activity of Prickly Pear Seeds Oil 

In vitro antioxidant activity of PPSO was determined using different methods, including DPPH, ABTS and the molybdate method, and the findings varied from one study to another study, mainly due to different analytical methods, solvent, and methods used for PPSO extraction.

Brahmi et al. [36] indicated that PPSO extracted by the cold press method showed antioxidant activity of 0.56 ± 0.01 AU, as determined using the molybdate method, and radical free radical scavenger ability of 37.0 ± 4.2%, as determined using the DPPH method. The reported antioxidant activity from this study is considered weak compared to another reported study [17], which showed that compared with vitamin C, PPSO extracted by cold pressing has 88% and 87% scavenging activity on DPPH (10-4M) and ABTS, respectively. In this regard, a study was performed by Berraaouan et al. [3], who showed that the Moroccan PPSO extracted by cold press exhibited a good antioxidant effect in DPPH scavenging assay (0.001%; *w*/*v*) with an IC_50_ value of 0.96 mg/mL. In this study, the antioxidant activity of PPSO extracted by ultrasound extraction ranged from 55.84 to 68.37 mg AAE/100 g for ABTS and 101.63 to 289.26 µmol TE/100 g for DPPH, which showed to be different from that extracted by the Soxhlet method with scavenging activity of 54.33 ± 0.84 mg AAE/100 g and 266.60 ± 1.97 µmol TE/100 g in ABTS and DPPH methods, respectively [11]. In addition, the antioxidant activity of PPSO extracted by supercritical fluid extraction showed an EC50 value of 140 mg extract/mg DPPH for inhibition of free radical formation, and that was better than the antioxidant activity of oil extracted by the Soxhlet method, which showed an EC50 value of 307 (mg extract/mg DPPH), as reported in [34].

Moreover, the antioxidant activity of PPSO is also influenced by the solvent used for extraction. The results of the free radical scavenging activity of PPSO extracted by different solvents, *n*-hexane, petroleum ether and chloroform-methanol (2:1, *v*/*v*), in comparison with antioxidant of vitamin E was reported [6]. The study clearly showed that the oil extracted using chloroform-methanol (2:1, *v*/*v*) exhibited higher antiradical activity (87%) towards the DPPH radical compared to *n*-hexane (86%) and petroleum ether (76%). In this regard, the antioxidant activity of PPSO from Algerian prickly pear seeds was reported. According to Chaalal et al. [43], the best results of antioxidant capacity were obtained when the seeds were extracted with 75% acetone (among ethanol, methanol, and water 50%, *v/v*), which had an antioxidant capacity of 95%.

Limited studies have reported on the antioxidant activity of PPSO in vivo. However, more recently, this was reported in [9]. (The findings obtained from their research showed that PPSO had significant antioxidant action in rats induced with acute inflammation, where the tested antioxidative parameters that advanced the oxidation protein product, including malondialdehyde, were significantly lower in treated rats with PPSO compared to untreated rats, and the level of these parameters returned to the normal level). Furthermore, PPSO has shown a significant evaluation of antioxidant enzyme activity compared to untreated rats, and the activity of superoxide dismutase, catalase and glutathione peroxidase reached the normal level with the group of animals that were not induced with inflammation [9].

### 5.2. Antimicrobial Activity of Prickly Pear Seeds Oil

Few studies have reported the antimicrobial activity of PPSO. In general, PPSO showed weak antimicrobial activity in some reported studies, no activity in other studies and good activity in other studies. 

Recently, the antimicrobial activity of Algerian PPSO at a concentration of 100 μL extracted by cold press was investigated [36]. Two Gram-positive strains (*Methicillin-resistant Staphylococcus aureus (MRSA) ATCC 43300* and *MRSA ATCC 29213*), two Gram-negative strains (*Escherichia coli* and *Pseudomonas aeruginosa*) and six fungal strains (*Aspergillus niger*, *Aspergillus flavus*, *Mucor rammaniarrus*, *Aspergillus ochraceus*, *Aspergillus parasitus* and *Candida albicans*) were selected to determine the antifungal activity. Findings from this study have indicated that the PPSO did not show antimicrobial activity against bacteria and fungi selected in this study. The authors have explained that the weak antimicrobial activity could have been due to the evaluation method used, which could influence the results, or the low content of phenolic compounds found in the studied oils, which could also be responsible for their inefficiency. However, a study of AbdelFattah et al. [12] indicated that PPSO extracted by methanol at 100 µL of 50% dilution (*v/v* methanol) reduced the growth of *Salmonella*, *Escherichia coli*, *Bacillus subtilis*, *Bacillus cereus*, yeast and mycotoxins metabolites while there is no significant effect response to yeast spores. In this regard, the antimicrobial activity of PPSO originated from Tunisia was tested against 10 microorganisms that have been known to be clinically pathogenic in humans; four bacterial strains, including *Escherichia coli*, *Staphylococcus aureus*, *Streptococcus agalactiae*, and *Enterobacter cloacae*, three yeast strains, including *Candida albicans*, *Candida parapsilosis*, and *Candida sake*, and three fungi, including *Aspergillus niger*, *Penicillium digitatum*, and *Fusarium oxysporum*, were studied [10]. Incubation with the PPSO at doses of 50 μL for 24 h for bacterial strains, 48 h for yeast, and 3–4 days for fungi at 28 °C indicated that PPSO showed antibacterial activity against *Enterobacter cloacae*, *Candida parapsilosis* and *Candida sake*, whereas no activity was shown against the three bacterial strains tested, including *Staphylococcus aureus*, *Streptococcus agalactiae*, and *Escherichia coli*. The oil also showed antifungal activity against three opportunistic cutaneous molds, including *Penicillium*, *Aspergillus*, and *Fusarium.* Moreover, the antimicrobial activity of the PPSO extracted from two Mexican varieties (green: *Opuntia albicarpa* and red: *Opuntia ficus indica*) with different solvents was reported [13]. In their study, the antimicrobial activity of PPSO was examined against different bacteria, including *Candida albicans*, *Escherichia coli*, *Staphylococcus aureus*, *Listeria monocytogenes*, *Pseudomonas aeruginosa*, *Saccharomyces cerevisiae* and *Salmonella Typhi*. Both PPSO from the two Mexican varieties produced a microbial inhibition zone in most of the microorganisms against Gram-positive and Gram-negative bacteria, and the effect was comparable to antimicrobial compounds such as ampicillin, streptomycin, and sulfamethoxazole/trimethoprim. 

It is well-known that the presence of biofilm in pathogenic microbes makes these pathogens hard to treat. In this regard, a study by Nazzaro et al. [45] evaluated the ability of PPSO at 1 to 8 μL/mL obtained through cold pressure to form biofilm using different types of bacteria, (including *Listeria monocytogenes*, *Escherichia coli*, *Pseudomonas aeruginosa*, and the phytopathogen *Pectobacterium carovotorum*). PPSO at 1 to 8 μL/mL was able to inhibit the biofilm of *Escherichia coli*, *Pseudomonas aeruginosa* and *Pectobacterium carotovorum* at 38.75%, 71.84% and 63.06% inhibition, respectively. The action of the PPSO was also 64.97% effective at blocking the metabolism of *Listeria monocytogenes* cells. When the PPSO was tested against *Pectobacterium carotovorum*, the microbial cell metabolism was completely inhibited with 8 μL/mL (96.26%). These findings indicate an interesting applicative versatility of this oil, with potentialities for food, agriculture and health purposes.

### 5.3. The Antidiabetic Properties of Prickly Pear Seeds Oil

The antidiabetic effect of PPSO is the most longstanding claimed pharmacological effect of PPSO. Thus far, most of the reported studies have addressed different mechanisms of action; however, the only scientific demonstration of a possible antidiabetic activity has been in rats.

Recently, in the study reported by AbdelFattah et al. [12], feeding male albino rats a diet supplemented with PPSO at 5% (*w*/*w*) for two months was stated to decrease the serum glucose concentration by 4.88 and 11.43% after 30 and 60 days, respectively, and to increase liver glycogen levels significantly by 53.94 and 125.85% compared to the control group. The suggested mechanism of action was due to increased insulin secretion, which stimulates glucose incorporation into glycogen in skeletal muscles and liver for the regulation of blood glucose. In addition, oral glucose test tolerance was performed on healthy Sprague Dawley rats. Oral administration of either aqueous or hydro-ethanol cactus seed extracts at 400 mg/kg for a period of 0.5–3 h, induced a reduction of blood glucose levels with a maximum decrease at an hour and a half compared to the control group by 62% [14]. Moreover, the antidiabetic effect of PPSO with its molecular mechanisms in streptozotocin (STZ)-induced diabetic rats was investigated [6]. Oral administration of PPSO at doses of 0.4 and 0.6 g/kg for 21 days to STZ rats caused a significant reduction of the blood glucose level compared to STZ untreated rats, and furthermore, the effect was approximately similar to the standard drug for blood glucose reduction, glibenclamide, at the end of the experiment period. The author has stated that treatment with PPSO elicited an increase in the expression level of glucose transporter 2 gene but reduced the expression of the phosphoenolpyruvate carboxykinase gene, which are key genes in the glucose metabolism. In this regard, the evaluation effect of PPSO treatment at 2 mL/kg of body weight on the incidence of alloxan-induced diabetic Swiss albino mice for seven days was studied [5]. Their results have shown that the administration of PPSO significantly decreased the incidence of alloxan-induced hyperglycemia in the PPSO-treated group compared to the untreated group. The authors suggested that this effect might be due to the synergism of the antioxidant compounds in quenching free radicals and the capacity of their unsaturated fatty acid in PPSO to enhance the antioxidant status in pancreatic b cells.

Moreover, an oral glucose test tolerance was performed on healthy or streptozotocin-induced diabetic rats as envisaged [3]. PPSO was administered to the rats at a dose of 0.8 mL/kg body weight, and glibenclamide was used as a standard drug at 2 mg/kg body weight. An evaluation of serum glucose level was carried out at 30, 60, 120, 240, and 360 min after treatment. The findings from their study indicated significant inhibition on the hyperglycemia that follows glucose loading at 90 min for treated rats. The author suggested that the mechanism of reduction of the glucose level could be due to inhibition of intestinal glucose absorption by the action of unsaturated fatty acid that presented in PPSO, where it has been known that fatty acids disturb the absorption function of enteric cells when they are present in the luminal space [46]. In another study, the effect of PPSO supplementation to healthy adult male rats of the Wistar strain at 25 mg/kg body weight for 2 months was investigated [15]. The obtained findings from this study indicated that PPSO caused a significant decrease in serum glucose concentration (22%), and that was parallel with the significant increase in liver glycogen levels as compared to the control group. The author has suggested the increase of glycogen levels in liver and muscle with the increase in insulin secretion, which stimulates glucose incorporation into glycogen in skeletal muscles and liver for the regulation of blood glucose.

### 5.4. Effect of Prickly Pear Seed Oil on Lipid Profile and Cholesterol Regulation

Hypercholesterolemia is associated with an increased risk of adverse cardiovascular outcomes. Limited studies reported the effect of PPSO on regulating the cholesterol and lipid profile, while there were several studies that reported the benefits of other parts of prickly pear, including fruits, leaves, pulp fractions and prickly pear pectin for cholesterol regulation and management of body weight in human and animal studies.

Evidence of the effect of PPSO in reducing the total cholesterol, LDL, and serum glucose levels in male Albino rats fed with a diet supplemented with PPSO at 5% (*w*/*w*) for two months compared to control animals was reported [12]. This study also indicated that PPSO had no significant effect on body weight gain but caused a decrease in feed conversion efficiency. The effect was shown to be due to the presence of different compounds in PPSO, including β-sitosterol and other compounds, such as unsaturated fatty acid, β-carotene and vitamin E, which have been reported in the literature to have a lowering effect on the lipid profile. In this regard, the modulation effect of Schottenol and Spinasterol, two sterols presented in PPSO as well as sterol extracts from PPSO on cholesterol metabolism in the Murine microglia BV2 cell line (BV2) at a concentration range from 12.5 to 50 μM for 24 and 48 h was reported [42]. PPSO modulated the gene expression of two nuclear receptors (liver X receptor (LXR)-α and LXRβ) and their target genes (ABCA1 and ABCG1), which have been known to be involved in regulating cholesterol metabolism. In addition, feeding adult male rats of Wistar strain a diet supplemented with PPSO (2.5%, *w*/*w*) for nine weeks resulted in a significant reduction in total cholesterol, triglyceride level, atherogenic index and average gain of body weight compared to the control group. The results from this study have also indicated that the fatty acid analysis in the liver of treated animals with PPSO showed that a significant increase in oleic acid levels resulted in an increase of MUFA in rats fed with a PPSO diet compared to other groups [47]. In this regard, Ennouri et al. [15] also indicated a reduction in plasma total cholesterol and LDL, VLDL cholesterol with no effect on HDL-cholesterol concentrations in male Wistar rats fed a diet supplemented with PPSO at 2.5% (*w*/*w*) for 2 months. This effect was suggested due to phytosterols presented in oil, especially β-sitosterol, where many studies have reported in the literature that phytosterols induce a decrease in lipoprotein cholesterol levels in total plasma.

### 5.5. Cytotoxic and Apoptotic Effects of Prickly Pear Seed Oil

A recent study [12] investigated the in vitro chemoprevention effect of PPSO at different concentrations (0.01, 0.1, 1, 10, 100 μM) against the growth of Colo-205 and HepG2 cells for 72 h. PPSO has potent activity (IC_50_ = 0.052 µM) toward HepG2, while this PPSO exhibited active but not potent activity (IC50 = 29.5 µM) against colorectal cancer (Colo-205) cell line. (The authors also found the inhibition effect of cell growth in cells that were treated with PPSO to be parallel with an increase of ROS in the cells, suggesting a ROS-induced cell death likely due to the pro-oxidant effects of the extracts). However, the PPSO was effective at inhibiting Colo-320 and Colo-741 growth [16], where the apoptotic effects of spiny (wild) and thornless (cultivated) forms of *Opuntia ficus indica* L. grown in Cyprus were studied against Colo-320 and Colo-741 cells for 48 h. Cell growth and cytotoxicity were measured by MTT assays. The spiny and thornless PPSO (1:16 dilution) were found to be active against the inhibition of Colo-320 and Colo-741 cell growth for 48 h. The author suggested that the inhibitory effect of thornless PPSO is due to the high linoleic acid content, which is a known compound with an anticancer effect in cancer cells. However, in some studies, the PPSO and its composition did not show a cytotoxic effect in vitro. In this regard, the sterol extract prepared from PPSO was screened for its cytotoxic effect against the Murine microglia BV2 cell line at a concentration range of 12.5–50 μM for 24 and 48 h using MTT assay; the findings indicated that sterol extract was not toxic to Murine microglial BV2 cells [48]. In addition, another study reported [49] that the prickly pear seed extracts obtained from different cultivars of prickly pear did not show any toxicity on breast, prostate and colon cancer cells in a concentration range of 0.2–0.16 g/mL assessed by the MTT assay, and it was toxic towards the MOLT-4 leukemia cells with an IC50 value of 5 mg/mL by one of the tested varieties.

### 5.6. Anti-Inflammatory Effect of Prickly Pear Seed Oil

Prickly pear seed oil showed a potential anti-inflammatory effect and reduced the inflammatory biomarkers in vivo, as reported recently [9]. In their study, acute inflammation was induced by carrageenan in adult male Wistar rats by administration of 100 μL of 1% freshly prepared solution of carrageenan in normal saline in the right hind paw of each rat. Rats were treated with PPSO at 25 μL/paw. The obtained results showed a significant reduction in the size of the edema in the oil-treated group, compared to all the studied groups, and the reduction effect was even better than the reference drug groups. PPSO also significantly reduced inflammatory biomarkers, including the number of white blood cells, blood platelets, C-reactive protein and plasma fibrinogen concentration compared to untreated animals. The author has explained that this effect is due to the antioxidant properties of PPSO and other bioactive compounds such as phytosterols, tocopherols, polyphenols, and carotenoids. In addition, PPSO was shown to have wound healing potential in vivo, as reported in [10]. The findings from this study have shown that the topical application of PPSO at 0.6 μL/mm^2^ accelerated skin closure and improved the healing process significantly compared to the reference and the control groups. The authors explained that the wound healing effect of the PPSO is due to the active components that are present in the oil, including unsaturated fatty acids, triacylglycerols, phytosterols, and tocopherols; these compounds work to enhance the speed of wound contraction, complete reepithelialization, and improve the external scar’s aspect. Moreover, Benattia et al. [14] investigated the anti-inflammatory effect of the prickly pear seed extract (aqueous and hydro-ethanolic) at a dose of 500 mg/kg in male Sprague Dawley rats. The seed extract showed a significant inhibition of the edema of paw in the rats induced by carrageenan with 25% inhibition for 3 h after pretreatment compared to untreated induced rats. The author suggested that the efficacy of the hydro-ethanolic extract from the seeds was due to the presence of polyphenolic compounds, among which flavonoids were reported to be able to inhibit the oxidants released by leukocytes and other phagocytes in the inflammatory site [5]. Flavonoids are also well-known to reduce the effect of prostaglandins, which cause the late phase of acute inflammation and pain perception [50].

### 5.7. Anti-Ulcer Activity of Prickly Pear Seed Oil

The only study that investigated the anti-ulcer activity of PPSO is [10]. The administration of PPSO at two different doses of 3.5 and 7 mL /kg/body to the male albino Wistar rats induced with 1% absolute ethanol was found to protect gastric mucosa against the ulcerating effect of ethanol. PPSO showed high efficiency in the protection of the cytoarchitecture and function of the gastric mucosa against the severe damages provoked by ethanol intake. PPSO showed healing of 91% on day 2 and 99% on day 3, and complete healing was attained on the fourth day under PPSO treatment. The anti-ulcer effect of PPSO was explained due to its richness in beneficial compounds, including tocopherols, linoleic acid, oleic acid, and β-sitosterol, which act in synergistic and complementary ways to ensure gastroprotection and gastric mucosal.

### 5.8. The Effect of Prickly Pear Seed Oil against UV-C Radiation

Even though a viable PPSO on the market is used mainly as a cosmetic ingredient, studies that have shown the activity of PPSO as a good cosmetic product are very limited. In this regard, the authors of [19] investigated the effect of PPSO on human dermic fibroblasts using a primary fibroblast culture of human skin. PPSO was applied at 10, 50, 100, and 200 µM, and then the culture cells were submitted to UV radiation in a crosslink for 15 min after that. PPSO at 50 µM reduced cell death due to UV radiation of human dermic fibroblasts. The authors explained that the protective effect of the PPSO against UV-C radiation was due to the action of the mixture of components that are present in PPSO against UV radiation, such as the antioxidants and polyunsaturated fatty acid. In addition, PPSO was suggested to be able to protect against UV-B and UV-A ranges, as reported in [26]. Findings from this study indicated that PPSO strongly absorbs UV-C radiation within a range of 100–290 nm and shows some absorbing properties in the UV-B range of 290–320 nm and UV-A ranges of 320–400 nm, where ultraviolet light is responsible for the most cellular damage.

### 5.9. Toxicity of Prickly Pear Seed Oil 

Few studies have investigated the acute toxicity of PPSO in animals. In [10], PPSO was administrated to the rats orally at two different doses, 3.5 and 7 mL/kg/body weight for five days. PPSO did not cause any toxicity symptoms or mortality in the animals with the two different doses during the 5 days of observation. In addition, acute toxicity in albino mice after oral or intraperitoneal administration of PPSO at doses of 1, 3 or 5 mL/kg was investigated by [5]. No mortality was observed, and there were no behavioral or autonomic effects throughout the 14-day period of observation in all treated mice, which showed that the PPSO is safe at doses up to 5 mL/kg. Moreover, another study [24] investigated the acute toxicity of PPSO in mice via the oral route with a single oral dose of 10, 20, 30, 40, 50, 60 or 70 mL/kg body weight and intraperitoneally with 0.5, 1, 2, 3, 4, 5 or 6 mL/kg body weight of PPSO. All doses of PPSO administered orally and intraperitoneally caused immediate agitation and behavioral perturbations with temporary writhing, followed by a quiet attitude period and sedation. Generally, diarrhea was observed, and the animals died 12 h after the administration of PPSO. The PPSO showed LD50 of 43 ± 0.8 mL/kg body weight in oral administration and LD50 of 2.72 ± 0.1 mL/kg body when applied intraperitoneally. These values are considered very high, which indicates the safety of PPSO.

## 6. Conclusions

The PPSO has been extracted from prickly pear seeds using different extraction techniques, from conventional to advanced, and the PPSO yield varied depending on many factors, including geographic region, harvest period, fruit variety, maturation, extraction method and type of extraction solvent. Based on physicochemical properties of PPSO, it is considered an edible oil and can be used by humans. The chemical characterization of the oil has been reported, and it is sufficiently understood that the PPSO has high nutritive value and can be further studied for its health promotion effects. PPSO is characterized by a high level of polyunsaturated fatty acids, with a high content of linoleic acid and a good level of oleic acid. PPSO oil in some varieties showed to contain high n-6 fatty acids. There has been a great deal of attention on n-3 fatty acids for health, but linoleic is now also being considered as beneficial. PPSO is also rich in phytosterols, among which β-sitosterol is the major. PPSO was found to be a good source of vitamin E, and γ-Tocopherol was found to be the major form of tocopherols in PPSO. PPSO contains carotenoids, phenolic compounds and several volatile compounds. Some differences in fatty acid composition and sterol composition were found to be due to different factors; mainly, location, varieties and methods of extraction. Experimental studies have reported that the PPSO shows several potential health benefits and biological activities, including in vivo and in vitro antioxidant activity, antimicrobial, antidiabetic, lipid-lowering activity and in vitro anticancer, anti-inflammatory, anti-ulcer activities; moreover, PPSO reduced cell death due to UV radiation of human dermic fibroblasts. Authors of a number of studies have explained that the reported benefits and activities are due to the compounds presented in the PPSO, mainly polyunsaturated fatty acids, sterols and compounds with antioxidant activity. However, the reported studies on biological activities did not sufficiently address the mechanisms of action to allow for the next phase of clinical trials or drug developments from the prickly pear seed oil.

## Figures and Tables

**Figure 1 molecules-26-05018-f001:**
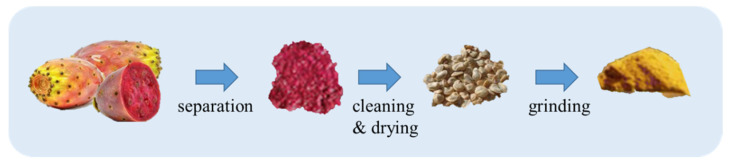
Preparation of prickly pear seeds for oil extraction [20,21].

**Figure 2 molecules-26-05018-f002:**
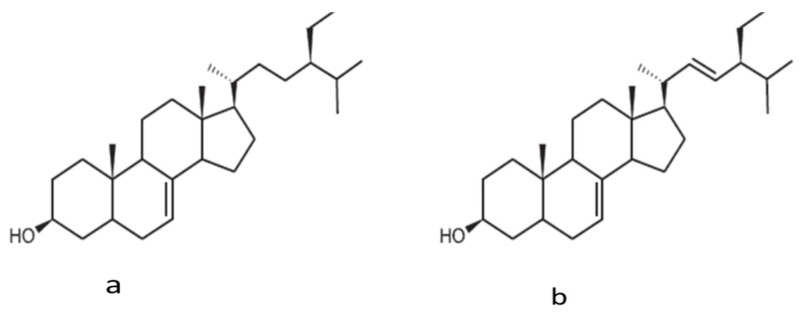
Schottenol (**a**) and spinasterol (**b**) structure presented in prickly pear seed oil. Sourced from El Kharrassi et al. [42].

**Figure 3 molecules-26-05018-f003:**
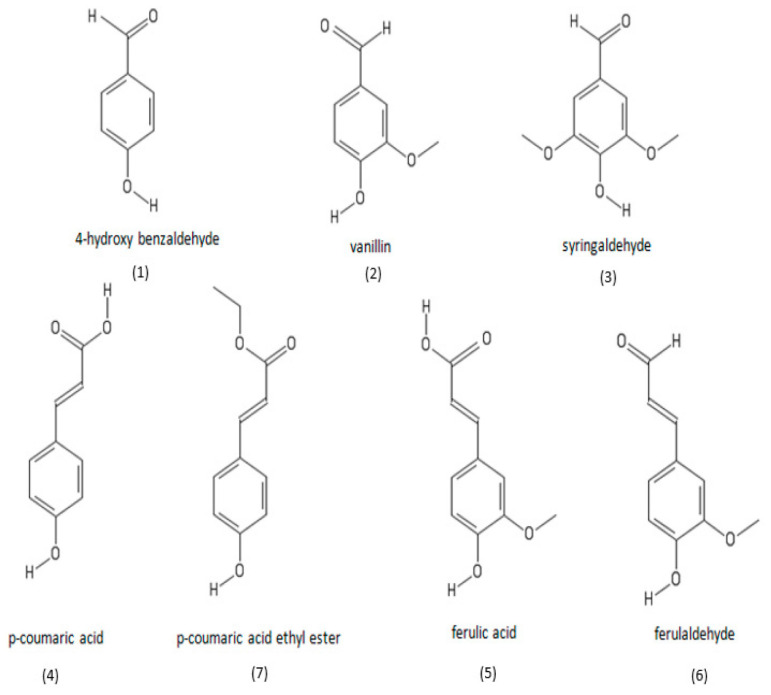
Structural formulas of phenolic compounds identified in prickly pear seed oil. Image from [44].

**Figure 4 molecules-26-05018-f004:**
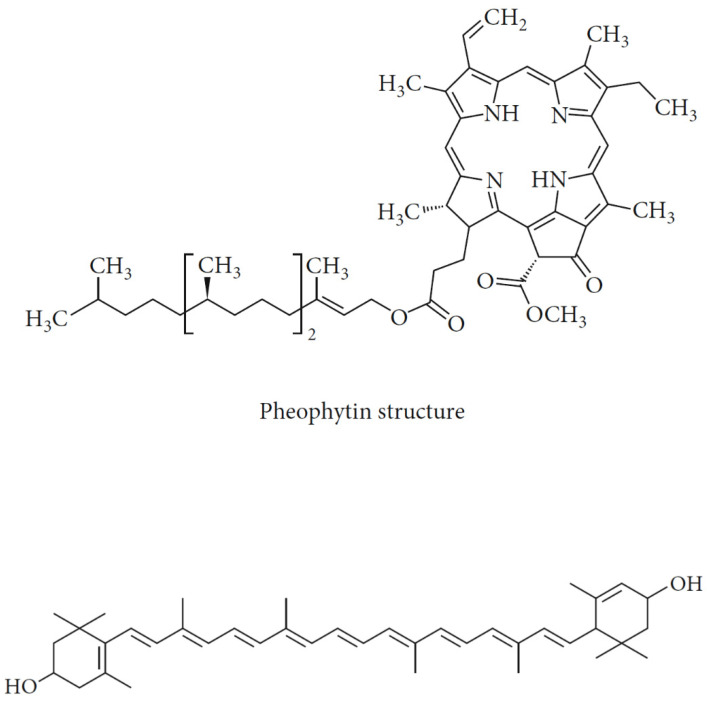
Chemical structures of pheophytin and lutein fractions in prickly pear seeds oil. Image from [17].

**Table 1 molecules-26-05018-t001:** Different extraction methods, solvent, extraction time and yield of prickly pear seed oil.

Extraction Technique	Method	Yield (%)	Reference
Solvent Extraction			
2-Methyloxolane and *n*-hexane	Soxhlet system for 8 h	9.55 ± 0.12 2-methyloxolane 8.86 ± 0.25 *n*-hexane	[22]
Diethyl ether	Soxhlet system for 3 h	2.61–7.69	[23]
Methanol	Soxhlet system for several h	16.20	[29]
*n*-Hexane	Soxhlet apparatus for 4–6 h	5.40 ± 0.5	[7]
*n*-Hexane	Soxhlet system	NR	[11]
*n*-Hexane	Soxhlet system for 8 h	5.40–9.9	[2]
*n*-Hexane	Soxhlet system for 18 h	10.45 ± 0.10	[25]
*n*-Hexane	Soxhlet system	8.91–10.32	[28]
*n*-Hexane	Soxhlet system for 9 h at 120 °C	7.30–9.3	[26]
*n*-Hexane	Soxhlet system for 10 h	11.75	[1]
*n*-Hexane	Soxhlet system for 9 h	11.00	[27]
Petroleum ether	Soxhlet system for 6 h at 50 °C	5.00–14.4	[8]
Petroleum ether	Soxhlet system	17.20 as total fat	[30]
Maceration Extraction with Solvents			
*n*-Hexane	Macerating for 24 h	6.2 ± 0.3–15.5 ± 0.5	[18]
Different solvents, *n*-hexane, ethanol, and ethyl acetate	Mixing the powdered seeds with solvents	hexane yield 11.8 ethanol and ethyl acetate 10.00	[13]
*n*-Hexane	Powdered seeds were immersed in hexane at 25 °C in a dark place for 24 h	NR	[11]
Chloroform and methanol 2:1	Powdered seeds were immersed in chloroform and methanol 2:1 while adding BHT at a concentration of 0.001%	4.1–8.8	[31]
Chloroform and methanol 2:1	Powdered seeds were immersed in chloroform and methanol 2:1 with adding BHT at a concentration of 0.001%	5.4–6.7	[32]
*n*-Hexane, petroleum ether and chloroform-methanol (2:1 *v*/*v*)	Maceration with solvents 1:5 (*w*/*v*): at room temperature for 24 h	Chloroform: methanol produced a yield of 7.8 ± 0.4 whereas, the yield extracted using hexane and petroleum ether were 5.0 ± 0.4 and 6.1 ± 0.4, respectively	[6]
Chloroform and methanol 2:1	Powdered seeds were immersed in chloroform and methanol 2:1	9.9	[4]
Mechanical Extraction			
Electric powered screw press	The extraction process was followed by centrifugation of the oil for 15 min at 3000 rpm to separate from the sediment	NR	[19]
Cold pressure	first cold pressure using a mechanical oil press	NR	[9]
Cold press methods	Compression of 20 t was applied on the piston by means of a hydraulic press	0.5–6.1	[18]
Cold pressing	first cold pressing using a mechanical machine	NR	[19]
Supercritical Fluid Extraction Method			
Supercritical fluid extraction method	Optimal supercritical antisolvent fractionation conditions were carried out at best conditions of 15 MPa pressure and 30 g min^−1^ CO_2_ flow rate	The recovery extract was rich in catechin, epicatechin, and ferulic acid	[34]
Supercritical fluid extraction method	Best conditions at pressure of 500 bar, temperature at 35 °C, extraction time 80 min and CO_2_ flow rate of 20 g/min	6.5	[33]
Supercritical fluid extraction method	Extraction was carried out at different temperatures and fixed pressure.	1.9 (Cultivated) 3.4 (wild)	[28]
Ultrasound extraction methods	Ultrasound at 1500 W with a constant frequency of 20 kHz and a probe of 25 mm for 5–15 min with a fixed temperature of 25 °C	3.8–6.0	[11]
Microextraction			
Microwave	Extraction process of PPSO was carried out by mixing about 20 mg of powder seeds with 1 mL of isooctane, then the mixture was put in a microwave oven at 100 °C, 150 w during 13 min, then filtered, then the oil further processed by transesterification and salinization	NR	[18]
Hydrodistillation using Clevenger-type apparatus	Ground seeds were extracted with *n*-hexane as a solvent and a Clevenger-type apparatus	0.6–3.0	[35]

NR = not reported.

**Table 2 molecules-26-05018-t002:** Physicochemical characteristics of prickly pear seed oil.

Reported Parameters	References				
	[22]	[36]	[17]	[25]	[26]
Physical state at room temperature	NR	NR	Liquid	Liquid	NR
Color	NR	NR	Greenish yellow	Yellow brown	NR
Odor	NR	NR	Slightly fruity	NR	NR
Property	NR	NR	Dry oil	NR	NR
Density at 20 °C (mass/volume)	NR	0.92 ± 0.01	0.91 ± 0.001	0.91 ± 0.01	0.90 ± 0.0
Refractive index at 20 °C	NR	1.47 ± 0.010	1.5 ± 0.001	1.48 ± 0.01	1.5 ± 0.0
Acid index	1.26 ± 0.5 3.02 ± 0.5	21.2 ± 0.5	1.95 ± 0.03	1.82 ± 0.01	NR
Peroxide index (meq O_2_/kg of oil)	3.5 ± 1.5 8.6 ± 2.5	12.0 ± 0.4	2.23 ± 0.06	NR	NR
Iodine index (g I_2_/100 g of oil)	131.5 ± 0.5 32 ± 0.3	NR	108.52 ± 0.25	93.45 ± 0.22	101.5 ± 1.0
Saponification index (mg of KOH/g oil)	NR	NR	171.40 ± 0.43	177.10 ± 0.05	169.0 ± 0.1
Extinction coefficient (K_232_)	2.8 ± 0.5 3.25 ± 0.5	0.08 ± 0.010	NR	NR	NR
K_270_	0.51 ± 0.5 2.11 ± 0.5	0.13 ± 0.020	NR	NR	NR

NR = not reported.

**Table 3 molecules-26-05018-t003:** Main fatty acid composition of prickly pear seed oil.

Source of PPSO	Main Fatty Acid						References
	Unsaturated fatty acid %				Saturated fatty acid %		
	Linoleic	Oleic	Linolenic	Vaccenic	Palmitic	Stearic	
Morocco	62.0	21.0	0.30	NR	12.0	3.0	[22]
Spanish	57.7–63.1	19.0–21.8	0.23–1.10	NR	12.5–15.1	2.6–4.1	[23]
Mexico	66.5–76.1 60.5–78.8	9.3–19.9 10.7–19.9	NR NR	NR NR	5.6–56.2 2.9–6.1	15.5–36.1 2.9–5.2	[18]
Algeria	49.7–56.1	15.6–19.3	0.24	4.30	10.1 ± 0.2	2.8 ± 0.2	[36]
Egypt	54.03	22.41	0.63	NR	17.11	3.49	[29]
Tunisia	61.6 ± 0.1	21.2 ± 0.16	0.20	NR	12.2 ± 0.03	3.3 ± 0.03	[17]
Cyprus	55.9–60.1	15.6–17.6	NR	NR	12.4	4.0	[16]
Sicilia/Italy	58.0	18.0	NR	6.3	12.0	4.0	[38]
South Africa	56.9–65.2	16.4–22.5	NR	NR	12.7–16.1	2.2–3.4	[31]
Yemen	57.0	22.30	0.23	NR	14.0	3.0	[6]
Morocco	60.2–64.6	18.2–22.3	0.3	NR	11.6–12.4	3.0–3.4	[2]
Tunisia	57.6–57. 5	22.3–25.3	0.2–0.5	NR	14.3–11.3	3.12–4.3	[28]
Turkey	49.3–62.1	13.0–23.5	0.3	5.0–6.3	10.6–12.8	3.3–5.4	[24]
Tunisia	70.3 ± 0.6	16.8 ± 0.5	NR	NR	9.3 ± 0.2	3.1 ± 0.1	[27]
Local market, Berlin, Germany	53.5 ± 4.9	18.3 ± 1.6	NR	2.6 ± 0.2	20.1 ± 2.3	2.7 ± 0.1	[4]

NR = not reported.

**Table 4 molecules-26-05018-t004:** Composition of phytosterols and vitamin E (%) of prickly pear seed oil.

Composition	Composition of Total Sterol and Total Vitamin E (%) References						
	[22]		[40]	[2]	[23]	[1]	[4]
	Hexane	2-MeO					
Phytosterol				NR		NR	
β-Sitosterol	75.6 ± 0.7	82.8 ± 0.7	74.7 ± 1.2	75.9–81.8	81.9	71.60	72.0
Campesterol	11.6 ± 0.1	12.3 ± 0.1	10.3 ± 0.1	8.9–13.1	6.4	20.2 ± 0.1	16.6 ± 0.2
Δ5-Avenasterol	4.4 ± 0.1	4.1 ± 0.1	5.1 ± 0.2	3.6–6.7	NR	5.1	NR
Stigmasterol	3.3 ± 0.03	3.4 ± 0.5	1.5 ± 0.04	1.8 –3.0	NR	4.7	3.0 ± 0.04
Δ7-Avenasterol	2.0 ± 0.03	2.2 ± 0.02	2.3 ± 0.1	0.1–0.9	NR	2.3 ± 0.01	2.9 ± 0.03
Δ7-Stigmasterol	1.8 ± 0.02	2.2 ± 0.02	1.1 ± 0.02	0.3–1.5	NR	-	0.5 ± 0.01
Cholesterol	1.8 ± 0.4	1.5 ± 0.4	0.97 ± 0.03	0.9–1.3	NR	0.1 ± 0.0	ND
Schottenol					1.29		
Spinasterol					1.60		
Campestanol					0.82		
Sitostanol					7.94	5.7 ± 0.02	
vitamin E							
α-tocopherol	1.8± 0.5	1.95 ± 0.5	1.1 ± 0.1	1.3–2.5	NR	0.11 ± 0.0	0.56
β-tocopherol	ND	ND	ND	0.0–0.3	NR	0.00	0.12
γ-tocopherol	68.4 ± 0.5	67.5 ± 0.5	98.5 ± 0.1	86.5–91.7	NR	94.1	80.0
δ-tocopherol	7.5 ± 0.5	8.26 ± 0.5	0.46 ± 0.1	6.5–10.9	NR	3.4	5.0

NR = not reported. ND = not detected.

**Table 5 molecules-26-05018-t005:** The main classes of volatile compounds of cactus seed oil reported in the literature.

Volatile Compound	Amount		
	References		
	[19]	[7]	[35]
Hexanal	57.4 mg/kg		
2-methyl propanal	38.9 mg/kg		
Acetaldehyde	16.2 mg/kg		
Acetic acid	10.9 mg/kg		
Acetoin	10.2 mg/kg		
2,3-butanedione	6.3 mg/kg		
Acids		2.70%	
Alcohols		9.13%	
Aldehydes		62.72%	
Esters		2.82%	
Hydrocarbons		5.06%	
Ketones		4.38%	
Hydrocarbons			38.5% in red variety
Fatty acids and derivatives			31.9% in red variety 68% in yellow variety
Terpenes			12.4% red and 10.9% yellow

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
