# Peer review of "Prickly Pear Seed Oil Extraction, Chemical Characterization and Potential Health Benefits"

_molecules, 2021, doi:10.3390/molecules26165018_

Round 1

Reviewer 1 Report

Prickly Pear Seed Oil Extraction, Chemical Characterization and Potential Health Benefits.

Opuntia is widely grown in arid and semi-arid regions of the world and considered an invasive species in some countries. The plant is extensively used for food and feed and the pads and fruit are a feature of Mexican cuisine. The seeds have been reported to contain only a low to moderate amount of oil. The manuscript focuses on this prickly pear seed oil (PPSO), reviewing chemical composition and potential health benefits. Surprisingly, the literature on PPSO is quite extensive. There has been a previous review on the composition and potential uses of opuntia seed oil (Ciriminna et al 2017 ref 39) but potential health benefits were not reviewed in detail ad the review is quite short. A more detailed review of the subject would therefore be justified.

The organization of the manuscript is good. Subject areas are clearly defined and broken down into logical sections. The tables with included references are useful as the text can be hard to read.

I would suggest a slight change to the Abstract to state that the oil is a high linoleic oil and composition is influenced by variety and environment, and also by the method of extraction.

Minor comments.

Do the authors have permission to use Figure 1 ?

Line 62-64 attempts to summarize extraction techniques but contains a mixture of extraction and process steps. The techniques in lines 62 to 64 are soxhlet extraction, cold press and supercritical CO2 extraction. Maceration, autoclave, microwave and untrasound are process steps used prior to the extraction, or during the extraction. A little clarification would be helpful.

Add a reference for line 69

Line 114. It is unclear why maceration is separate, maceration or grinding is usually part of any solvent extraction process.

Line 153 and ref 33. This describes the extraction of residue after the seeds had already been extracted using a cold pressing method. The focus was not on oil content or yield. Move this sentence to the end of the section as it does not fit where it is.

Line 153. There are additional references describing the supercritical CO2 extraction of seed oil from opuntia. Some could be listed, although it is not necessary to discuss each in detail. For example Liu et al 2009 Food Chemistry 114:334-339.

Table 1. Oil density. The value from ref 24 is questionable. Density of 2.04 is too high, oil usually floats on water !

Line 380, remove the , after 81

Line 391 what is 5.12-stigmastadienol

Line 399, remove the single f

Line 403 add units for 14.44

Line 416 Chlorophyll has two a double l at the end

Line 496 sterols not setrols

Line 546 What species is methicillin resistant?

Line 362 what is parrell ?

Line 695, acute rather than cute….

Line 705 and others. Be consistent in use of italics for in-vivo and in-vitro.

The abstract and conclusion appropriately reflect that it is hard to draw any real conclusions on the health attributes of the oil, given the diverse and variable reports that are in the literature.

One thing that may be worth adding is that the oil is high in n-6 fatty acids. There has been a great deal of attention on n-3 fatty acids for health, but linoleic is now also being considered as beneficial (see Marangoni et al 2020 Atherosclerosis 292:90-98 for example).

Reviewer 2 Report

This study systematically reviews the related research of prickly pear seed oil (PPSO). Including extraction methods, chemical properties and biological activity evaluation. Here are some suggestions:

  1. The authors discuss the extraction method of prickly pear seed oil in detail. However, A table summary the extraction parameters is necessary including extraction method, solvent, extraction time and yield, etc.
  2. The 232 and 270 in table 1 for extinction coefficient need to be subscripted.
  3. For DPPH test, the concentration of PPSO in test solution need to be mentioned
  4. Table 4 is missing.

Reviewer 3 Report

The manuscript “
 Prickly Pear Seed Oil Extraction, Chemical Characterization and 2 Potential Health Benefits”
Was well written and cover extensive literature from 
 Opuntia ficus-indica L.)

The review describes the extraction by conventional and innovative methods (supercritical fluid, microwave extraction, and ultrasound extraction.
The fatty acids and polyphenols (catechin, epicatechin, and ferulic acid) Fatty acid composition as well phytosterols were described.
Prickly pear fatty acids and polyphenols are possibly related to the anti-inflammatory and antioxidants properties of the plant. 
The content of the phytosterols and tocopherols of prickly pear seed oil were described too and point the phytosterol as a fingerprint of the oil and it can be used for authenticity or detecting adulterations. The text highlighted β-sitosterol and campesterol that were found to be the major phytosterols compounds.

As the most medicinal plants and functional plant foods a series of benefits were described: antioxidant, antimicrobial, Anti-inflammatory antidiabetic, and, Anti-ulcer activities; cholesterol regulation, cytotoxic and apoptotic effects. 

Finally, the authors conclude that the “reported studies on biological activities did not sufficiently address the mechanisms of action to allow for the next phase of clinical trials or drug developments from the prickly pear seeds oil”.

There are two minor points to be review:

Figure 4 needs the best quality image to be published.
Line 787: In the conclusion: maybe need a correction changing "addressed" to "address".

I think that the review covers an interesting area of Prickly Pear Seed Oil and can be published in the Molecules.
